

# Mapping of soil properties at high resolution in Switzerland using boosted geoadditive models

Madlene Nussbaum[1], Lorenz Walthert[2], Marielle Fraefel[2], Lucie Greiner[3], and Andreas Papritz[1]

[1]Institute of Biogeochemistry and Pollutant Dynamics, ETH Zurich, Universitätstrasse 16, CH-8092 Zürich, Switzerland
[2]Swiss Federal Institute for Forest, Snow and Landscape Research (WSL), Zürcherstrasse 111, CH-8903 Birmensdorf, Switzerland
[3]Research Station Agroscope Reckenholz-Taenikon ART, Reckenholzstrasse 191, CH-8046 Zürich, Switzerland

*Correspondence to:* M. Nussbaum (madlene.nussbaum@env.ethz.ch)

**Abstract.**

High-resolution maps of soil properties are a prerequisite for assessing soil threats and soil functions and to foster sustainable use of soil resources. For many regions in the world precise maps of soil properties are missing, but often sparsely sampled and discontinuous (legacy) soil data are available. Soil property data (response) can then be related by digital soil mapping (DSM) to spatially exhaustive environmental data that describe soil forming factors (covariates) to create spatially continuous maps. With air- and spaceborne remote sensing data and multi-scale terrain analysis large sets of covariates have become common. Building parsimonious models, amenable to pedological interpretation, is then a challenging task.

We propose a new boosted geoadditive modelling framework (geoGAM) for DSM. A geoGAM models smooth nonlinear relations between responses and single covariates and combines these model terms additively. Residual spatial autocorrelation is captured by a smooth function of spatial coordinates and nonstationary effects are included by interactions between covariates and smooth spatial functions. The core of fully automated model building for geoGAM is componentwise gradient boosting.

We illustrate the application of the geoGAM framework by using soil data from the Canton of Zurich, Switzerland. We modelled effective cation exchange capacity (ECEC) in forest topsoils as continuous response. For agricultural land we predicted the presence of waterlogged horizons in given soil depth layers as binary and drainage classes as ordinal responses. For the latter we used proportional odds geoGAM taking the ordering of the response properly into account. Fitted geoGAM contained only few covariates (7 to 17) selected from large sets (333 covariates for forests, 498 for agricultural land). Model sparsity allowed covariate interpretation by partial effects plots. Prediction intervals were computed by model-based bootstrapping for ECEC. Predictive performance of the fitted geoGAM, tested with independent validation data and specific skill scores (SS) for continuous, binary and ordinal responses, compared well with other studies that modelled similar soil properties. SS of 0.23 up to 0.53 (with SS = 1 for perfect predictions and SS = 0 for zero explained variance) were achieved depending on response and type of score. geoGAM combines efficient model building from large sets of covariates with ease of effect interpretation and therefore likely raises the acceptance of DSM products by end-users.



# 1 Introduction

Soils fulfil many functions important for agriculture, forestry and the management of soil resources and natural hazards. The functionality of soils depends on their properties, hence, accurate and spatially highly resolved maps of basic soil properties such as texture, organic carbon content and pH for defined soil depth are needed for sustainable management of soils (FAO

and ITPS, 2015). Unfortunately, such soil property maps are often missing and availability of soil information is very different between nations and continents (Omuto and Nachtergaele, 2013). For areas where spatially referenced, but discontinuous and sparse (legacy) soil data is available, e.g. soil datasets consisting of soil profile data and laboratory measurements, these point data can be linked using digital soil mapping (DSM) techniques (e.g. McBratney et al., 2003; Scull et al., 2003) to spatial information on soil formation factors to generate spatially continuous maps.

In the past, many DSM approaches have been proposed to exploit the correlation between soil properties (response $Y(\mathbf{s})$) and soil forming factors (covariates $\mathbf{x}(\mathbf{s})$). *Linear regression modelling* (LM, see McBratney et al., 2003, for DSM applications) and *kriging with external drift* (EDK), its extension for autocorrelated errors (Bourennane et al., 1996; Nussbaum et al., 2014), have been often used. Strengths of LM and EDK are the ease of interpretation of the fitted models (e.g. by partial residual plots, Faraway, 2005, p. 73). This is important for checking whether modelled relations between the target soil property and

soil forming factors accord with pedological expertise and for conveying results of DSM analyses to users of such products. LM and EDK capture only linear relations between the covariates and a response. By using interactions between covariates, one can sometimes account for nonlinear relationships, but this quickly becomes unwieldy for a large number of covariates (e.g. above 30). Fitting models to (very) large sets of covariates has become common with the advent of remotely sensed data (Ben-Dor et al., 2009; Mulder et al., 2011) and novel approaches for terrain analysis (Behrens et al., 2010). Model building, i.e.

covariate selection, is then a formidable task. Although specialized methods like L2-boosting (Bühlmann and Hothorn, 2007) and lasso (Hastie et al., 2009, chap. 3) are available, they have not often been used for DSM (Nussbaum et al., 2014; Liddicoat et al., 2015; Fitzpatrick et al., 2016). *Generalized linear models* (GLM, e.g. Dobson, 2002) extend linear modelling to binary, nominal (e.g. soil taxonomic units) or ordinal responses (e.g. soil drainage classes). Although GLM are nonlinear models, the nonlinearly transformed conditional expectation $g(\mathrm{E}[Y(\mathbf{s})|\mathbf{x}(\mathbf{s})]) - g(\cdot)$ is some known link function – still depends linearly

on covariates.

Lately, *tree-based machine learning methods* have become popular for DSM: Classification and regression trees (CART, see references in McBratney et al., 2003), *Cubist*, (e.g. Henderson et al., 2005; Adhikari et al., 2013; Lacoste et al., 2016) and ensemble tree methods like *random forest* (RF, e.g. Grimm et al., 2008; Wiesmeier et al., 2011) and *boosted trees*, (BRT, e.g. Moran and Bui, 2002; Martin et al., 2011) were used. All tree-based methods easily account for complex nonlinear relations

between responses and covariates. They model continuous and categorical responses (albeit without making a difference between nominal and ordinal responses), inherently deal with incomplete covariate data and allow to model spatially changing (nonstationary) relationships. BRT and RF fit models to large sets of covariates. The structure of the fitted models can be explored by variable importance and partial dependence plots (Hastie et al., 2009, Sect. 10.9, and Martin et al., 2011, for an



application). Nevertheless, tree-based ensemble methods remain complex, and results are not as easy to interpret regarding the relevant soil forming factors of a case study as results from (G)LM.

*Generalized additive models* (GAM, e.g. Hastie and Tibshirani, 1990, Chapt. 6) offer a compromise between ease of interpretation and flexibility in modelling nonlinear relationships. GAM expand the (possibly transformed) conditional expectation

of a response given covariates as an additive series

$$g\left(\mathrm{E}[Y(\mathbf{s})\,|\,\mathbf{x}(\mathbf{s})]\right) = \nu + f(\mathbf{x}(\mathbf{s})) = \nu + \sum_j f_j(x_j(\mathbf{s})), \tag{1}$$

where $\nu$ is a constant and $f_j(x_j(\mathbf{s}))$ are linear terms or unspecified "smooth" nonlinear functions of single covariates $x_j(\mathbf{s})$ (e.g. smoothing spline, kernel or any other scatterplot smoother) and $g(\cdot)$ is again a link function. GAM extend GLM to account for truly nonlinear relations between $Y$ and $\mathbf{x}$ (and not just for nonlinearities imposed by $g$), but they limit the complexity of

the fitted functions to additive combinations of simple nonlinear terms and thereby avoid the curse of dimensionality (Hastie et al., 2009, Sect. 2.5). For continuous, ordinal and nominal responses, GAM can be readily fitted to large sets of covariates by boosting (Hofner et al., 2014; Hothorn et al., 2015). Boosting handles covariate selection and avoids over-fitting if stopped early (Bühlmann and Hothorn, 2007). Hence, the structure of boosted GAM can be more easily checked and interpreted than RF and BRT models. GAM have occasionally been used for DSM but were never very popular (see references in McBratney et al.,

2003). Recently, Poggio et al. (2013) and Poggio and Gimona (2014) used GAM to model continuous and binary responses.

Besides precise predictions, sometimes also accurate modelling of *prediction uncertainty* matters for DSM studies (e.g. for mapping temporal changes of soil carbon and nutrients stocks). *Quantile regression forest* (Meinshausen, 2006), an extension of RF, estimates the quantiles of the distributions $Y(\mathbf{s})|\mathbf{x}(\mathbf{s})$ and provides prediction intervals directly. Prediction intervals can also easily be constructed for predictions by (G)LM and GAM, as long as the uncertainty arising from model building is

ignored. To take the effect of model building properly into account one resorts best to *bootstrapping* (Davison and Hinkley, 1997, Sect. 6.3.3). Bootstrapping is also useful to model prediction uncertainty for boosted models, which *per se* do not qualify the precision of predictions, and to account for all sources of prediction uncertainty of regression kriging approaches (Viscarra Rossel et al., 2014).

In summary, a versatile DSM procedure should

1. model nonlinear relations between $Y(\mathbf{s})$ and $\mathbf{x}(\mathbf{s})$, where responses and covariates may be continous, binary, nominal or ordinal variables,

2. efficiently build models with good predictive performance for large sets of covariates ($p >> 30$),

3. preferably result in parsimonious models with a simple structure that can be easily interpreted and checked for plausibility, and

4. accurately quantify the precision of predictions computed from the fitted models.

The objective of our work was to develop a DSM framework that optimizes requirements 1–4 based on *boosted geoadditive models* (geoGAM), an extension of GAM for spatial data. First, we introduce the modelling framework and describe in detail





the model building procedure. Second, we apply the method to three DSM case studies from the Canton of Zurich, Switzerland, aiming at different types of responses: Effective cation exchange capacity (ECEC) of forest topsoils (continuous response), presence/absence of morphological features for waterlogging in agricultural soils (binary response), and drainage classes, characterizing prevalence of anoxic conditions, again in agricultural soils (ordinal response). To assess the validity of the

modelling results with independent data (obtained by splitting the original dataset into calibration and validations subsets), we used specific criteria that take the nature of the various responses properly into account. These criteria are in common use for forecast verification in atmospheric sciences (e.g. Wilks, 2011), but, to our knowledge, have not been much used for (cross-)validating DSM predictions.

## 2  geoGAM modelling framework

### 2.1  Model representation

A generalized additive model (GAM) is based on the following components (Hastie and Tibshirani, 1990, Chapt. 6 and Eq. (1)):
i) *Response distribution*: Given $\mathbf{x}(\mathbf{s}) = x_1(\mathbf{s}), x_2(\mathbf{s}),$
$..., x_p(\mathbf{s})$, the $Y(\mathbf{s})$ are conditionally independent observations from simple exponential family distributions. ii) *Link function*:
$g(\cdot)$ relates the expectation $\mu(\mathbf{x}(\mathbf{s})) = \mathrm{E}[Y(\mathbf{s})|\mathbf{x}(\mathbf{s})]$ of the response distribution to iii) the *additive predictor*

$\sum_j f_j(x_j(\mathbf{s}))$.

geoGAM extend GAM by allowing a more complex form of the additive predictor (Kneib et al., 2009; Hothorn et al., 2011): First, one can add a smooth function $f_{\mathbf{s}}(\mathbf{s})$ of the spatial coordinates (smooth spatial surface) to the additive predictor to account for residual autocorrelation. More complex relationships between $Y$ and $\mathbf{x}$ can be modelled by adding terms like $f_j(x_j(\mathbf{s})) \cdot f_k(x_k(\mathbf{s}))$ – capturing the effect of interactions between covariates – and $f_{\mathbf{s}}(\mathbf{s}) \cdot f_j(x_k(\mathbf{s}))$ – accounting for spatially

changing dependence between $Y$ and $\mathbf{x}$. Hence, in its full generality, a generalized additive model for spatial data is represented by

$$g(\mu(\mathbf{x}(\mathbf{s}))) = \nu + f(\mathbf{x}(\mathbf{s})) \ =$$
$$\nu + \underbrace{\sum_u f_{j_u}(x_{j_u}(\mathbf{s})) + \sum_v f_{j_v}(x_{j_v}(\mathbf{s})) \cdot f_{k_v}(x_{k_v}(\mathbf{s}))}_{\text{global marginal and interaction effects}}$$
$$+ \underbrace{\sum_w f_{\mathbf{s}_w}(\mathbf{s}) \cdot f_{j_w}(x_{j_w}(\mathbf{s}))}_{\text{nonstationary effects}} + \underbrace{f_{\mathbf{s}}(\mathbf{s})}_{\text{autocorrelation}} \ . \tag{2}$$

Kneib et al. (2009) called Eq. (2) a geoadditive model, a name coined before by Kammann and Wand (2003) for a combination of Eq. (1) with a geostatistical error model.

It remains to specify what response distributions and link functions should be used for the various response types: For (possibly transformed) *continuous* responses one uses often a normal response distribution combined with the identity link





$g\left(\mu(\mathbf{x(s)})\right) = \mu(\mathbf{x(s)})$. For binary data (coded as 0 and 1), one assumes a Bernoulli distribution and uses often a logit link

$$g\left(\mu(\mathbf{x(s)})\right) = \log\left(\frac{\mu(\mathbf{x(s)})}{1 - \mu(\mathbf{x(s)})}\right), \tag{3}$$

where

$$\mu(\mathbf{x(s)}) = \mathrm{Prob}[Y(\mathbf{s}) = 1 \,|\, \mathbf{x(s)}] = \frac{\exp(\nu + f(\mathbf{x(s)}))}{1 + \exp(\nu + f(\mathbf{x(s)}))}. \tag{4}$$

For ordinal data, with ordered response levels, $1, 2, \ldots, k$, we used the cumulative logit or proportional odds model (Tutz, 2012, Sect. 9.1). For any given level $r \in (1, 2, \ldots, k)$, the logarithm of the odds of the event $Y(\mathbf{s}) \leq r \,|\, \mathbf{x(s)}$ is then modelled by

$$\log\left(\frac{\mathrm{Prob}[Y(\mathbf{s}) \leq r \,|\, \mathbf{x(s)}]}{\mathrm{Prob}[Y(\mathbf{s}) > r \,|\, \mathbf{x(s)}]}\right) = \nu_r + f(\mathbf{x(s)}), \tag{5}$$

with $\nu_r$ a sequence of level-specific constants satisfying $\nu_1 \leq \nu_2 \leq \ldots \leq \nu_r$. Conversely,

$$\mathrm{Prob}[Y(\mathbf{s}) \leq r \,|\, \mathbf{x(s)}] = \frac{\exp(\nu_r + f(\mathbf{x(s)}))}{1 + \exp(\nu_r + f(\mathbf{x(s)}))}. \tag{6}$$

Note that $\mathrm{Prob}[Y(\mathbf{s}) \leq r \,|\, \mathbf{x(s)}]$ depends on $r$ only through the constant $\nu_r$. Hence, the ratio of the odds of two events $Y(\mathbf{s}) \leq r \,|\, \mathbf{x(s)}$ and $Y(\mathbf{s}) \leq r \,|\, \tilde{\mathbf{x}}(\mathbf{s})$ is the same for all $r$ (Tutz, 2012, p. 245).

### 2.2   Model building (selection of covariates)

To build parsimonious models that can readily be checked for agreement with pedological understanding, we applied a number of fully automated steps 1–6. In several of these steps we optimized tuning parameters by 10-fold cross-validation with

fixed subsets using either root mean squared error (RMSE, Eq. (12), continuous responses), Brier score (BS, Eq. (16), binary responses) or ranked probability score (RPS, Eq. (18), ordinal responses) as optimization criteria. To improve the stability of the algorithm continuous covariates were first scaled (by difference of maximum and minimum value) and centred.

    1. Boosting (see step 2 below) is more stable and converges more quickly when the effects of categorical covariates (factors) are accounted for as model offset. We therefore used the group lasso (least absolute shrinkage and selection operator,

Breheny and Huang, 2015) – an algorithm that likely excludes non-relevant covariates and treats factors as groups – to select important factors for the offset. For ordinal responses (Eq. (6)) we used stepwise proportional odds logistic regression in both directions with BIC (e.g. Faraway, 2005, p. 126) to select the offset covariates because lasso cannot be used for such responses.

    2. Next, we selected a subset of relevant factors, continuous covariates and spatial effects by componentwise gradient

boosting. Boosting is a slow stagewise additive learning algorithm. It expands $f(\mathbf{x(s)})$ in a set of base procedures (baselearners) and approximates the additive predictor by a finite sum of them as follows (Bühlmann and Hothorn, 2007):

      (a) Initialize $\hat{f}(\mathbf{x(s)})^{[m]}$ with offset of step 1 above and set $m = 0$.





(b) Increase $m$ by 1. Compute the negative gradient vector $\mathbf{U}^{[m]}$ (e.g. residuals) for a loss function $l(\cdot)$.

(c) Fit all baselearners $g(\mathbf{x}(\mathbf{s}))_{1..p}$ to $\mathbf{U}^{[m]}$ and select the baselearner, say $g(\mathbf{x}(\mathbf{s}))_j^{[m]}$ that minimizes $l(\cdot)$.

(d) Update $\hat{f}(\mathbf{x}(\mathbf{s}))^{[m]} = \hat{f}(\mathbf{x}(\mathbf{s}))^{[m-1]} + v \cdot g(\mathbf{x}(\mathbf{s}))_j^{[m]}$ with step size $v \leq 1$.

(e) Iterate steps (b) to (d) until $m = m_{\text{stop}}$ (main tuning parameter).

We used the following settings in above algorithm: As loss functions $l(\cdot)$ we used $L_2$ for continuous, negative binomial likelihood for binary (Bühlmann and Hothorn, 2007) and proportional odds likelihood for ordinal responses (Schmid et al., 2011). Early stopping of the boosting algorithm was achieved by determining optimal $m_{\text{stop}}$ by cross-validation. We used default step length ($v = 0.1$). This is not a sensitive parameter as long as it is clearly below 1 (Hofner et al., 2014). For continuous covariates we used penalized smoothing spline baselearners (Kneib et al., 2009). Factors were treated as linear baselearners. To capture residual autocorrelation we added a bivariate tensor-product P-spline of spatial coordinates (Wood, 2006, pp. 162) to the additive predictor. Spatially varying effects were modelled by baselearners formed by multiplication of continuous covariates with tensor-product P-splines of spatial coordinates (Wood, 2006, pp. 168). Uneven degree of freedom of baselearners biases baselearner selection (Hofner et al., 2011). We therefore penalized each baselearner to 5 degrees of freedom ($df$). Factors with less than 6 levels ($df < 5$) were aggregated to grouped baselearners. By using an offset, effects of important factors with more than 6 levels were implicitly accounted for without penalization.

3. At $m_{\text{stop}}$ (see step 2 above), many included baselearners had very small effects only. To remove these we computed the effect size $e_j$ of each baselearner $f_j(x_j(\mathbf{s}))$. For factors the effect size $e_j$ was the largest difference between effects of two levels and for continuous covariates it was equal to the maximum contrast of estimated partial effects (after removal of extreme values as in boxplots, Frigge et al., 1989). We fitted generalized additive models (GAM, Wood, 2011) by including smooth and factor effects depending on the effect size $e_j$ of the corresponding baselearner $j$. We iterated through $e_j$ and excluded covariates with $e_j$ smaller than a threshold effect size $e_t$. Optimal $e_t$ was determined by 10-fold cross-validation of GAM. In these GAM fits smooth effects were penalized to 5 degrees of freedom as imposed by componentwise gradient boosting (step 2 above). The factors selected as offset in step 1 were now included in the main GAM, that was fitted without offset.

4. We further reduced the GAM by stepwise removal of covariates by cross-validation. The candidate covariate to drop was chosen by largest $p$ value of $F$ tests for linear factors and approximate $F$ test (Wood, 2011) for smooth terms.

5. Factor levels with similar estimated effects were merged stepwise again by cross-validation based on largest $p$ values from two sample $t$-tests of partial residuals.

6. The final model (used to compute spatial predictions) was a parsimonious GAM. Because of step 5, factors had possibly a reduced number of coefficients. Effects of continuous covariates were modelled by smooth functions and – if at all present – spatially structured residual variation (autocorrelation) was represented by a smooth spatial surface. To avoid over-fitting both types of smooth effects were penalized to 5 degrees of freedom (as imposed by step 2).





Model building steps 1 to 6 were implemented in the R package `geoGAM` (Nussbaum, 2017).

### 2.3 Predictions and predictive distribution

Soil properties were predicted for new locations $\mathbf{s}_+$ from the final geoGAM fit by $\tilde{Y}(\mathbf{s}_+) = \hat{f}(\mathbf{x}(\mathbf{s}_+))$. To model the predictive distributions for continuous responses we used a non-parametric, model-based bootstrapping approach (Davison and Hinkley, 1997, pp. 262, 285) as follows:

A. New values of the response were simulated according to $Y(\mathbf{s})^* = \hat{f}(\mathbf{x}(\mathbf{s})) + \epsilon$, where $\hat{f}(\mathbf{x}(\mathbf{s}))$ are the fitted values of the final model and $\epsilon$ are errors randomly sampled with replacement from the centred, homoscedastic residuals of the final model (Wood, 2006, p. 129).

B. The geoGAM was fitted to $Y(\mathbf{s})^*$ according to steps 1–6 of Sect. 2.2.

C. Prediction errors were computed according to $\delta_+^* = \hat{f}(\mathbf{x}(\mathbf{s}_+))^* - (\hat{f}(\mathbf{x}(\mathbf{s}_+)) + \epsilon)$ , where $\hat{f}(\mathbf{x}(\mathbf{s}_+))^*$ are predicted values at new locations $\mathbf{s}_+$ of the model built with the simulated response $Y(\mathbf{s})^*$ in step B above, and the errors $\epsilon$ are again randomly sampled from the centred, homoscedastic residuals of the final model (see step A).

Prediction intervals were computed according to

$$[\hat{f}(\mathbf{x}(\mathbf{s}_+)) - \delta_{+(1-\alpha)}^* ; \, \hat{f}(\mathbf{x}(\mathbf{s}_+)) - \delta_{+(\alpha)}^*]. \tag{7}$$

where $\delta_{+(\alpha)}^*$ and $\delta_{+(1-\alpha)}^*$ are the $\alpha$- and $(1-\alpha)$-quantiles of $\delta_+^*$, pooled over all 1000 bootstrap repetitions.

Predictive distributions for binary and ordinal responses were directly obtained from a final geoGAM fit by predicting probabilities of occurrence $\widetilde{\mathrm{Prob}}(Y(\mathbf{s}) = r \,|\, \mathbf{x}(\mathbf{s}))$ (Davison and Hinkley, 1997, p. 358).

## 3 Case studies - Materials and Methods

### 3.1 Study regions

We applied the modelling framework to 3 datasets on properties of forest and agricultural soils in the Canton of Zurich in Switzerland (Fig. 1). Forests (ZH forest), as defined by the Swiss topographic landscape model (swissTLM3D, Swisstopo, 2013a), cover an area of 506.5 $\mathrm{km}^2$, or roughly 30 % of the total area of the Canton of Zurich. The spatial extent of the agricultural region was chosen near the Lake Greifensee by the availability of imaging spectroscopy data collected by the APEX spectrometer (Schaepman et al., 2015). Agricultural land was defined as the area not covered by any areal features such as settlements or forests extracted from the Swiss topographic landscape model (swissTLM3D, Swisstopo, 2013a). Wetlands, forests, parks or city gardens were excluded, resulting in a study region of 170 $\mathrm{km}^2$.

In the Canton of Zurich, forests extend across altitudes ranging from 340 to 1170 m above sea level (a.s.l), and the Greifensee area elevation ranges from 390 to 840 m a.s.l. (Swisstopo, 2016). The climatic conditions (period 1961–1990, Zimmermann

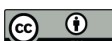



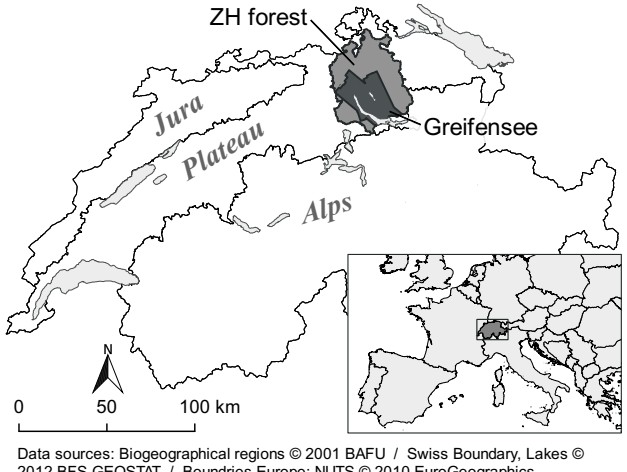

Data sources: Biogeographical regions © 2001 BAFU / Swiss Boundary, Lakes © 2012 BFS GEOSTAT / Boundries Europe: NUTS © 2010 EuroGeographics

**Figure 1.** Location of the study regions Greifensee and ZH forest on the Swiss Plateau.

and Kienast, 1999) vary accordingly, with mean annual rainfall between 880–1780 mm for the forested and 1040–1590 mm for the agricultural study region. Mean annual temperatures range between 6.1–9.1 °C and 7.5–9.1 °C, respectively. Two thirds of the forested area is dominated by coniferous trees (FSO, 2000b). Half of the Greifensee study region is covered by crop land and one third by permanent grassland. The remainder are orchards, horticultural areas or mountain pastures (Hotz et al.,

2005). In the Canton of Zurich, soils formed mostly from Molasse formations and quaternary sediments dominantly from the last glaciation (Würm). In the north-eastern part, the Jura foothills with limestone rocks reach into the ZH forest study region (Hantke, 1967).

## 3.2   Data

### 3.2.1   Soil data base

We used legacy soil data collected between 1985 and 2014. Data originates from long-term soil monitoring of the Canton of Zurich (KaBo), a soil pollutant survey (Wegelin, 1989), field surveys for creating soil maps of the agricultural land (Jäggli et al., 1998) or soil investigations in the course of different projects by the Swiss Federal Institute for Forest, Snow and Landscape Research (WSL, Walthert et al., 2004). Sites for pollutant surveying were chosen on a regular grid, those for creating soil maps were determined purposively by field surveyors to best represent soils typical for the given landform. The sites of WSL were

chosen purposively according to the aims of the project. Collating these soil data from different sources implicated that soil data were not directly comparable, and tailored harmonisation procedures were required to provide consistent soil datasets. The heterogeneity of soil legacy data resulted among others from several standards of soil description and soil classification, different data keys, different analytical methods and in particular, often missing metadata for a proper interpretation of the datasets. Therefore, we elaborated a general harmonisation scheme that covers performance steps required to merge different





soil legacy data into one common consistent database (Walthert et al., 2016). Sampling sites were recorded in the field on topographic maps (scale 1:25 000), hence accuracy of coordinates is about $\pm$ 25 m.

### 3.2.2 Effective cation exchange capacity (ECEC, forest soils)

After the removal of sites with missing covariate values, we used 1844 topsoil samples from 1348 sites with data on effective

cation exchange capacity (ECEC). Most measurements refer to composite samples where aliquots were measured in 20 by 20 m squares from 0–20 cm soil depth. At about 100 sites soil profiles were sampled at genetic horizons. ECEC [$\mathrm{mmol_c\,kg^{-1}}$] for 0–20 cm was computed from horizon data by

$$\mathrm{ECEC}_{0-20} = \sum_{i=1}^{h} w_i\, \mathrm{ECEC}_i, \tag{8}$$

where $\mathrm{ECEC}_i$ is the value for horizon $i$, $w_i$ is a weight given by soil density $\rho_i$ and the fraction of the thickness of horizon

$i$ within 0–20 cm and $h$ is the number of horizons intersecting the 0–20 cm layer. The $w_i$ were normalized to sum to 1. $\rho_i$ was estimated from soil organic matter (SOM) and/or sampling depth by a pedotransfer function (PTF see Supplement of Nussbaum et al., in prep.). Due to a lack of respective data, the volumetric stone content was assumed to be constant.

For most soil samples, ECEC was measured after extraction in an ammonium chloride solution (FAC, 1989; Walthert et al., 2004, 2013). Roughly 5 % of the samples had only measurements of Ca, Mg, K and Al (extracted by ammonium acetate EDTA

solution, Lakanen and Erviö, 1971; ELF, 1996; Gasser et al., 2011). For these samples, we estimated ECEC by using a PTF (Nussbaum and Papritz, 2015).

We assigned 293 of 1348 sites (528 samples) to the validation set, which was used to check the predictive performance of the fitted statistical model, and the remaining 1055 sites (1316 samples) were used to calibrate the model. The legacy samples were spatially clustered. To ensure that the validation sites were evenly spread over the study region, the validation sites were

selected by weighted random sampling. The weight attributed to a site was proportional to the forested area within its Dirichlet polygon (Dirichlet, 1850).

We found a considerable variation in ECEC values ranging from 17.4 to 780 $\mathrm{mmol_c\,kg^{-1}}$ (median 141.1 $\mathrm{mmol_c\,kg^{-1}}$, Table S1 in Supplement). On average, ECEC was slightly larger in the calibration than in the validation set.

### 3.2.3 Presence of waterlogged soil horizons (agricultural soils)

Waterlogging characteristics were recorded in the field at 962 sites within the Greifensee study region by visual evaluation (Jäggli et al., 1998). Swiss soil classification distinguishes horizon qualifiers *gg* (strongly gleyic, predominantly oxidized) and *r* (anoxic, predominantly reduced) and both are believed to limit plant growth (Jäggli et al., 1998; Müller et al., 2007; Litz, 1998; Danner et al., 2003; Kreuzwieser and Rennberg, 2014).

We constructed binary responses for three soil depth layers 0–30 cm, 0–50 cm and 0–100 cm. If one of the horizon qualifiers

*gg* or *r* was recorded within the interval, we assigned 1 = *presence of waterlogged horizons* and 0 = *absence of waterlogged soil horizons* otherwise.



We chose 198 of 962 sites to form a validation set, again by using weighted random sampling. The remaining 764 sites were used to build and fit the models. In the topsoil (0–30 cm) *gg* or *r* horizon qualifiers were only observed at 13.4 % of the 962 sites. Down to 50 cm about twice as many sites (25.9 %) showed signs of anoxic conditions and down to 1 m already 38.6 % of sites featured an anoxic or gleyic horizon (Table S2 in Supplement).

### 3.2.4 Drainage classes (agricultural soils)

Swiss soil classification differentiates hydromorphic features of soils in more detail describing the degree, depth and source of waterlogging by 3 supplementary qualifiers for stagnic, gleyic or anoxic profiles (I, G, R; categorical attributes, Brunner et al., 1997). To reduce complexity of classification, we aggregated these qualifiers to three ordered levels *well drained* (qualifiers I1–I2, G1–G3, R1 or no hydromorphic qualifier), *moderately well drained* (I3–I4, G4) and *poorly drained* (G5–G6, R2–R5).

For validation we used the same 198 sites as for *presence of waterlogged soil horizons*, but only 732 sites were used for model building due to missing data in the covariates. The majority (66.6 %) of the 930 sites were *well drained*, only 12.7 % were classified as *moderately well drained* and 20.7 % as *poorly drained* (Table S3 in Supplement).

### 3.2.5 Covariates for statistical modelling

To represent local soil formation conditions, we used data from 23 sources (Table 1). For ECEC a total of 333 covariates were used describing climatic (71 covariates) and topographic conditions (196 covariates). For the agricultural land, we used in addition 180 spectral bands of the APEX spectrometer, spatial information on historic wetlands and agricultural drainage networks resulting in 498 covariates in total.

### 3.3 Statistical analysis

We built models for the five responses according to Sect. 2.2 and computed predictions for new locations at nodes of a 20 m-grid. Predictions were post-processed in the following way:

### 3.3.1 Response transformation

ECEC data in 0-20 cm soil depth was positively skewed (Table S1 in Supplement), hence we fitted the model to the log-transformed data. In full analogy to lognormal kriging (Cressie, 2006, Eq. (20)), the predictions were backtransformed by

$$E[Y(\mathbf{s})\,|\,\mathbf{x}] = \exp\left(\hat{f}(\mathbf{x}(\mathbf{s})) + \frac{1}{2}\hat{\sigma}^2 - \frac{1}{2}\mathrm{Var}[\hat{f}(\mathbf{x}(\mathbf{s}))]\right) \tag{9}$$

with $\hat{f}(\mathbf{x}(\mathbf{s}))$ being the prediction of the log-transformed response, $\hat{\sigma}^2$ the estimated residual variance of the final geoGAM fit and $\mathrm{Var}[\hat{f}(\mathbf{x}(\mathbf{s}))]$ the variance of $\hat{f}(\mathbf{x}(\mathbf{s}))$ as provided again by the final geoGAM. Limits of prediction intervals were back-transformed by $\exp(\cdot)$ as they are quantiles of the predictive distributions.



**Table 1.** Overview of geodata and derived covariates, for more information see Supplement of Nussbaum et al. (in prep.) ($r$: pixel resolution for raster datasets or scale for vector datasets, $a$: only available for study region Greifensee (Gr) or ZH forest (Zf), NDVI: normalized differenced vegetation index, TPI: topographic position index, TWI: topographic wetness index, MRVBF: multi-resolution valley bottom flatness).

| geodata set | $r$ | $a$ | covariate examples |
|---|---|---|---|
| **Soil** | | | physiographical units, historic wetland |
| Soil overview map (FSO, 2000a) | 1:200 000 | | presence, presence of drainage |
| Wetlands Wild maps (ALN, 2002) | 1:50 000 | Gr | networks or soil ameliorations |
| Wetlands Siegfried maps (Wüst-Galley et al., 2015) | 1:25 000 | Gr | |
| Anthropogenic soil interventions (AWEL, 2012) | 1:5 000 | Gr | |
| Drainage networks (ALN, 2014b) | 1:5 000 | Gr | |
| **Parent material** | | | (aggregated) geological units, ice level |
| Last Glacial Maximum (Swisstopo, 2009) | 1:500 000 | | during last glaciation, presence of |
| Geotechnical map (BFS, 2001) | 1:200 000 | | aquifer |
| Geological map (ALN, 2014a) | 1:50 000 | | |
| Groundwater occurrence (AWEL, 2014) | 1:25 000 | Gr | |
| **Climate** | | | mean annual/monthly temperature, |
| MeteoSwiss 1961–1990 (Zimmermann and Kienast, 1999) | 25/100 m | | precipitation, radiation, degree days, |
| MeteoTest 1975–2010 (Remund et al., 2011) | 250 m | | $NH_3$ concentration in air |
| Air pollutants (BAFU, 2011) | 500 m | Zf | |
| $NO_2$ immissions (AWEL, 2015) | 100 m | Gr | |
| **Vegetation** | | | band ratios, NDVI, 180 hyperspectral |
| Landsat7 scene (USGS EROS, 2013) | 30 m | | bands, aggregated vegetation units, |
| DMC mosaic (DMC, 2015) | 22 m | | canopy height |
| SPOT5 mosaic (Mathys and Kellenberger, 2009) | 10 m | Zf | |
| APEX spectrometer mosaics (Schaepman et al., 2015) | 2 m | Gr | |
| Share of coniferous trees (FSO, 2000b) | 25 m | Zf | |
| Vegetation map (Schmider et al., 1993) | 1:5 000 | Zf | |
| Species composition data (Brassel and Lischke, 2001) | 25 m | Zf | |
| Digital surface model (Swisstopo, 2011) | 2 m | Zf | |
| **Topography** | | | slope, curvature, northness, TPI, TWI, |
| Digital elevation model (Swisstopo, 2011) | 25 m | | MRVBF (various radii/resolutions) |
| Digital terrain model (Swisstopo, 2013b) | 2 m | | |



### 3.3.2 Conversion of probabilistic to categorical predictions

For binary and ordinal responses, Eq. (4) and (6) predict probabilities of the respective response levels. To predict the "most likely" outcome one has to apply a threshold to these probabilities. For binary data we predicted *presence of waterlogged horizons* if the probability exceeded the optimal value of the Gilbert skill score (GSS, Sect. 3.3.3) that discriminated *presence* and *absence of waterlogged horizons* best in cross-validation of the final geoGAM. GSS was selected because *absence of waterlogged horizons* was more common than *presence*, especially in topsoil. To ensure consistency of maps for sequential soil depth layers we assigned *presence of waterlogged horizons* to the lower depth layer if it was predicted for the layer above.

For ordinal responses we predicted the level to which the median of the probability distribution $\widetilde{\mathrm{Prob}}(Y(\mathbf{s}) \leq r | \mathbf{x}(\mathbf{s}))$ was assigned (Tutz, 2012, p. 475).

### 3.3.3 Evaluating the predictive performance of the statistical models

The predictive performance of the geoGAM, fitted for the continuous response ECEC, was tested by comparing predictions $\tilde{Y}(\mathbf{s}_i)$ (Eq. (9)) with measurements $Y(\mathbf{s}_i)$. Marginal bias and overall precision were assessed by

$$\mathrm{BIAS} = -\frac{1}{n}\sum_{i=1}^{n}(Y(\mathbf{s}_i) - \tilde{Y}(\mathbf{s}_i)), \tag{10}$$

$$\mathrm{robBIAS} = -\mathrm{median}_{1 \leq i \leq n}\left(Y(\mathbf{s}_i) - \tilde{Y}(\mathbf{s}_i)\right), \tag{11}$$

$$\mathrm{RMSE} = \left(\frac{1}{n}\sum_{i=1}^{n}\left(Y(\mathbf{s}_i) - \tilde{Y}(\mathbf{s}_i)\right)^2\right)^{1/2}, \tag{12}$$

$$\mathrm{robRMSE} = \mathrm{MAD}_{1 \leq i \leq n}\left(Y(\mathbf{s}_i) - \tilde{Y}(\mathbf{s}_i)\right), \tag{13}$$

$$\mathrm{SS_{mse}} = 1 - \frac{\sum_{i=1}^{n}\left(Y(\mathbf{s}_i) - \tilde{Y}(\mathbf{s}_i)\right)^2}{\sum_{i=1}^{n}\left(Y(\mathbf{s}_i) - \frac{1}{n}\sum_{i=1}^{n}Y(\mathbf{s}_i)\right)^2}, \tag{14}$$

where MAD is the median absolute deviation. $\mathrm{SS_{mse}}$ was defined as mean squared error skill score (Wilks, 2011, p. 359) with the sample mean of the measurements as reference prediction method. Interpretation is similar to $R^2$ with $\mathrm{SS_{mse}} = 1$ for perfect predictions and $\mathrm{SS_{mse}} = 0$ for zero explained variance. $\mathrm{SS_{mse}}$ becomes negative if the root mean squared error (RMSE) exceeds the standard deviation of the data. To validate the accuracy of the bootstrapped predictive distributions we plotted the empirical distribution function of the probability integral transform (Wilks, 2011, p. 375), which is equivalent to a plot of the coverage of one-sided prediction intervals $(0, \tilde{q}_\alpha(\mathbf{s}))$ against the nominal probabilities $\alpha$ used to construct the quantiles $\tilde{q}_\alpha(\mathbf{s})$.

For binary responses the predictive performance of fitted geoGAM was evaluated by the Brier skill score (BSS, Wilks, 2011, Eq. (8.37))

$$\mathrm{BSS} = 1 - \frac{\mathrm{BS}}{\mathrm{BS_{ref}}} \tag{15}$$



where the Brier score (BS) is computed by

$$\text{BS} = \frac{1}{n}\sum_{i=1}^{n}(y_i - o_i)^2. \tag{16}$$

where $n$ is the number of sites, $y_i = \widetilde{\text{Prob}}[Y(s_j) = 1 \,|\, x(s_j)]$ are the predicted probabilities and $o_j = \text{I}(Y(s_j) = 1)$ the observation. $\text{BS}_{\text{ref}}$ is the BS of a reference prediction where always the more abundant level (*absence of waterlogged horizons*)

is predicted. After transforming the predicted probabilities to the binary levels *presence or absence of waterlogged horizons* (Sect. 3.3.2) we further evaluated the bias ratio, Peirce skill score (PSS) and GSS. Bias ratio is the ratio of the number of *presence* predictions to the number of *presence* observations (Wilks, 2011, Eq. (8.10)). PSS is a skill score based on the proportion of correct *presence* and *absence* predictions where the reference predictions are purely random predictions that are constrained to be unbiased (Wilks, 2011, Eq. (8.16)). GSS is a skill score that uses the threat score as precision measure

(Wilks, 2011, Eq. (8.18)) and again random predictions as reference. Perfect predictions have PSS and GSS equal to 1, for random predictions the scores are equal to 0 and predictions worse than the reference receive negative scores. PSS is truly and GSS asymptotically equitable, meaning that purely random and constant predictions get the same scores (see Wilks, 2011, p. 316 and 321 for details).

For the ordinal response drainage classes we tested the fitted geoGAM by evaluating the ranked probability skill score

(RPSS), computed analogously to BSS by

$$\text{RPSS} = 1 - \frac{\text{RPS}}{\text{RPS}_{\text{ref}}} \tag{17}$$

where RPS is the ranked probability score (RPS, Wilks, 2011, Eq. (8.52)) given by

$$\text{RPS} = \sum_{i=1}^{n}\sum_{j=1}^{J}(Y_{i,j} - O_{i,j})^2 \tag{18}$$

with $Y_{i,j} = \widetilde{\text{Prob}}[Y(s_i) \le j \,|\, x(s_i)]$ being the predicted cumulative probabilities up to class $j$ and $O_{i,j} = \sum_{r=1}^{j} I(Y(s_i) = r)$

indicating observed absence (0) or presence (1) up to class $j$. $\text{RPS}_{\text{ref}}$ is the RPS for a reference that predicts always the most abundant class (*well drained*). For predictions of the ordinal outcomes (Sect. 3.3.2) we also computed the mean bias ratio and two skill scores: We calculated the mean bias ratio from three bias ratios created analogously to the binary case. These two-class settings were achieved by stepwise aggregation of two out of three classes (*well* vs. *moderately well* or *poorly drained*, then *well* or *moderately well* vs. *poorly drained* etc., Wilks, 2011, p. 319). PSS was computed in its general form (Wilks,

2011, p. 319) together with the Gerrity score (GS), which applies weights to the joint distribution of predicted and observed classes to consider their ordering and frequency (Wilks, 2011, p. 322).

### 3.3.4   Software

Terrain attributes were computed by ArcGIS (version 10.2, ESRI, 2010) and SAGA 2.1.4 (version 2.1.4, Conrad et al., 2015). All statistical computations were done in R (version 3.2.2, R Core Team, 2016) using several add-on packages, in particular





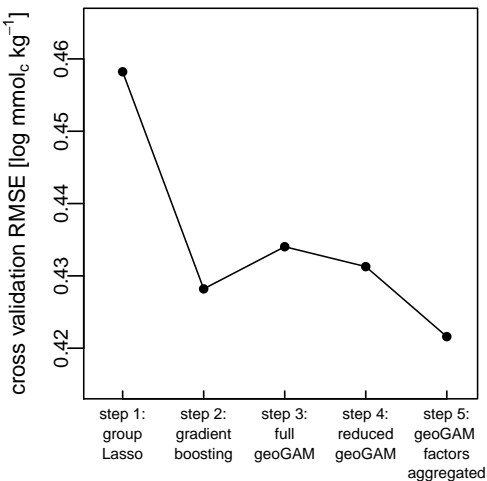

**Figure 2.** Change of cross-validation root mean squared error (RMSE) in steps 1–5 of model building procedure (Sect. 2.2).

`grpreg` for group lasso (version 2.8-1, Breheny and Huang, 2015), `MASS` for proportional odds logit regression (version 7.3-43, Venables and Ripley, 2002), `mboost` for componentwise gradient boosting (version 2.5-0, Hothorn et al., 2015), `mgcv` for geoadditive model fits (version 1.8-6, Wood, 2011), `raster` for spatial data processing (version 2.4-15, Hijmans et al., 2015) and `geoGAM` for the model building routine (version 0.1-2, Nussbaum, 2017).

## 4   Results

### 4.1   ECEC – case study 1

#### 4.1.1   Models for ECEC in 0-20 cm depth

Figure 2 shows the change of RMSE during model building (10-fold cross-validation). The small root mean squared error (RMSE) of 0.428 log $\mathrm{mmol_c\,kg^{-1}}$ after the gradient boosting step – with coefficients shrunken by the algorithm – could further be reduced (RMSE 0.422 log $\mathrm{mmol_c\,kg^{-1}}$) by removing covariates and by factor aggregation. Aggregating factor levels resembles shrinking of coefficients of such covariates.

Starting with 333 covariates model building successfully reduced the number of covariates in the model to 17. The remaining ones characterized geology, vegetation and topography (Table 2). Effective cation exchange capacity (ECEC) depended nonlinearly on nearly all continuous covariates, but nonlinearities were in general rather weak. (Fig. S1 in Supplement). No $f_s(\mathbf{s})$ term was included in the model, because residual autocorrelation was very weak (Fig. S2 in Supplement). Including nonstationary effects in the model would have improved the model only slightly (RMSE 0.406 log $\mathrm{mmol_c\,kg^{-1}}$), but would have added considerable complexity to the final model (21 covariates including 8 interactions with $f_s(\mathbf{s})$ terms).



**Table 2.** Covariates contained in final geoGAM for responses ECEC, *presence of waterlogged horizons* and drainage classes. More details can be found in Fig. S1 and S4 to S6 in Supplement . (*p*: number of covariates, SD: standard deviation in local neighbourhood, TPI: topographic position index, TWI: topographic wetness index, MRVBF: multiresolution valley bottom flatness).

| | ECEC 0-20 cm | presence of waterlogged horizons down to | | | drainage class |
| | | 30 cm | 50 cm | 100 cm | |
| --- | --- | --- | --- | --- | --- |
| *p* | 17 | 7 | 12 | 14 | 11 |
| Legacy soil data | correction factor | | | | |
| Geology, land use | distance to moraines, aquifer map, overview soil map, geological map, geotechnical map | historic wetlands | historic wetlands, drainage systems map | historic wetlands, drainage systems map, anthropogenic soil disturbance, extent last glaciation, geological map | historic wetlands, drainage systems map, aquifer map |
| Climate | — | global radiation, precipitation | global radiation, precipitation | dew point temperature | precipitation |
| Vegetation | SPOT5 vegetation index, vegetation map | — | UK-DMC green band | — | UK-DMC green band |
| Topography | SD slope, northness, ruggedness, surface convexity, negative openness, vertical distance to rivers | curvature, eastness, roughness, negative openness | SD elevation, SD slope, curvature, negative openness, TPI, TWI, MRVBF | SD elevation, curvature, eastness, convergence index, terrain texture, horizontal distance to rivers, TWI, MRVBF | SD elevation, terrain texture, TPI, TWI, MRVBF |

### 4.1.2 Validation of predicted ECEC with independent data

Predictive performance, as evaluated at 293 independent validation sites, was satisfactory. Figure 3 shows for the validation set measured ECEC in 0–20 cm plotted against the predictions. The solid line of the loess scatterplot smoother (Cleveland, 1979) is close to the 1:1 line indicating absence of conditional bias. This was confirmed by small marginal BIAS measures (Table 3). BIAS$^2$-to-MSE ratio was small for both log-transformed and original data (1.2 and 0.7 %, respectively). robRMSE




**Table 3.** Validation statistics for (a) log-transformed and (b) backtransformed ECEC 0–20 cm [$mmol_c\,kg^{-1}$] calculated for 528 samples (293 sites) of the validation set (definition of statistics see Sect. 3.3.3).

|  | BIAS | robBIAS | RMSE | robRMSE | $SS_{mse}$ |
|---|---|---|---|---|---|
| (a) | 0.052 | 0.006 | 0.471 | 0.411 | 0.407 |
| (b) | 6.3 | 8.9 | 74.9 | 55.3 | 0.365 |

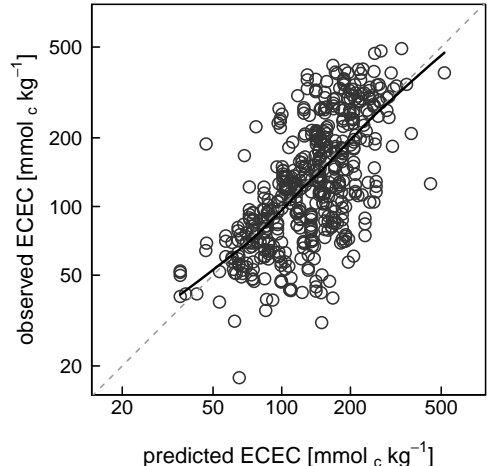

**Figure 3.** Scatter plot of measured against predicted ECEC in 0-20 cm mineral soil depth, computed with geoGAM (Sect. 4.1.1) for the sites of the validation set (solid line: loess scatter plot smoother, n: number of measurements).

(0.411 log $mmol_c\,kg^{-1}$) was somewhat smaller than RMSE (0.471 log $mmol_c\,kg^{-1}$) indicating that a few outlying ECEC observations were not particularly well predicted. RMSE of backtransformed data (74.9 $mmol_c\,kg^{-1}$) was also larger than its robust counterpart robRMSE (55.3 $mmol_c\,kg^{-1}$). The model explained about 40 % of the variance of the log-transformed and 37 % of the variance of the original data.

5    Figure 4 shows somewhat too large coverage for quantiles in the lower tails of the predictive distributions, hence the extent of lower tails of bootstrapped predictive distributions was underestimated. Upper tails of the predictive distributions were modelled accurately as the coverage was close to the nominal probabilities there. The coverage of symmetric 90 %-prediction intervals was again too small (84.1 %) because the lower tails were too short. The median width of 90 %-prediction intervals was equal to 201.8 $mmol_c\,kg^{-1}$, demonstrating that prediction uncertainty remained substantial, in spite of $SS_{mse}$ of nearly
10    40 %.




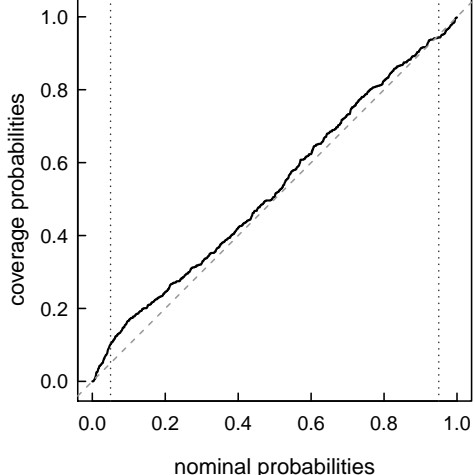

**Figure 4.** Coverage of one-sided bootstrapped prediction intervals $(0, q_\alpha(s_i))$ for 528 ECEC validation samples, plotted against nominal probability $\alpha$ used to construct the upper limit $q_\alpha$ of the prediction intervals (Vertical lines mark the 5 and 95 % probabilities).

### 4.1.3 Mapping ECEC for ZH forest topsoils

Predictions of ECEC were computed by the final geoGAM for the nodes of a 20 m-grid (Fig. 5). 44 % of the mapped topsoil has large to very large ECEC values. In contrast, 13 % ($\sim$66 km$^2$) of the forest topsoils in the study region are acidic with ECEC below 100 mmol$_c$ kg$^{-1}$. These soils are mostly found in the northern part of the Canton of Zurich. The spatial pattern of the width of 90 %-prediction intervals (Fig. S3 in Supplement) and of the mean predictions (Fig. 5) was very similar (Pearson correlation = 0.981), which follows from the lognormal model that we adopted for this response.

### 4.2 Presence of waterlogged soil horizons – case study 2

### 4.2.1 Models for presence of waterlogged horizons

Not surprisingly, the models for *presence of waterlogged horizons* in the three soil depths contained similar covariates, characterizing mostly wet soil conditions such as historic wetland maps, a map of agricultural drainage systems or several climatic covariates (Table 2). The same terrain attributes were repeatedly chosen for the three depths (Figs. S4 to S6 in Supplement). For all three depths model selection resulted in parsimonious sets of only 7 to 14 covariates chosen from a total of 498 covariates. The Brier skill score (BSS), computed using 10-fold cross-validation, increased from 0.350 for the 0–30 cm layer to 0.704 for the 0-100 cm layer suggesting that *presence of waterlogged horizons* can be better modelled when they occur more frequently. Degree of residual spatial autocorrelation on logit-scale was stronger in the 0–30 cm than in 0-100 cm layer (Fig. S2 in Supplement) confirming that the model performed better for the 0–100 cm layer. Adding a $f_s(\mathbf{s})$ term did not improve cross-validated





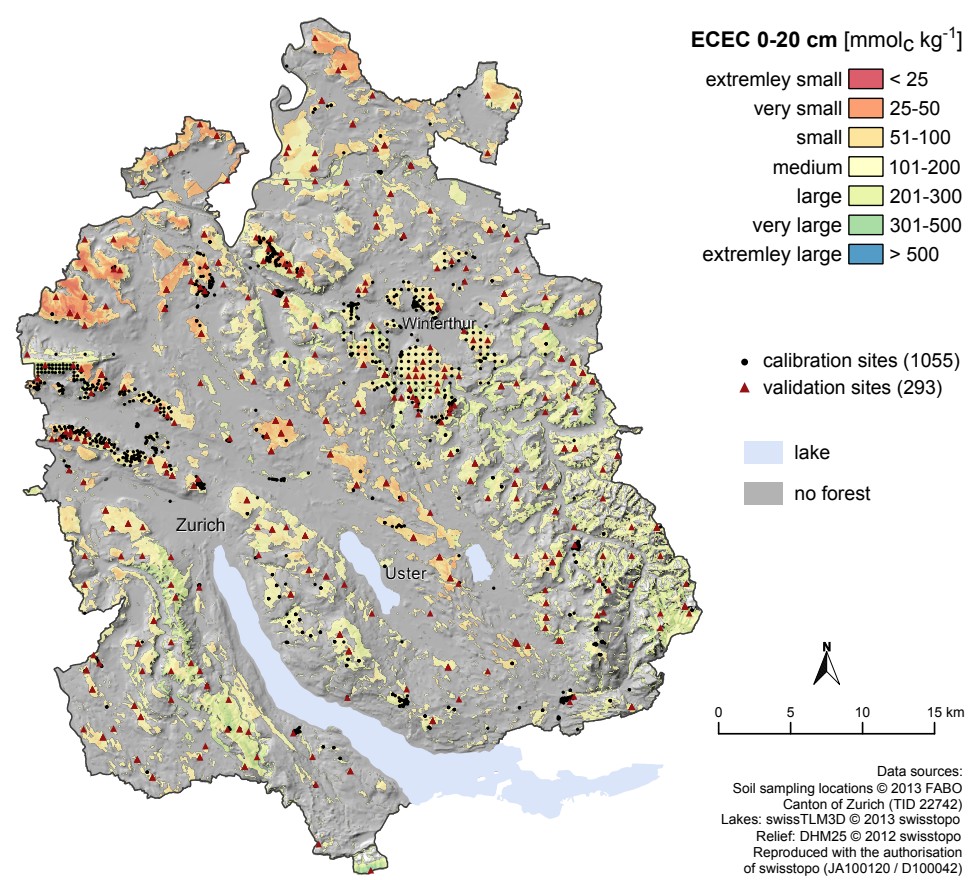

**Figure 5.** geoGAM predictions of effective cation exchange capacity (ECEC) in 0-20 cm depth of the mineral soil of forests in the Canton of Zurich, Switzerland (computed on a 20 m-grid with final geoGAM with covariates according to Table 2. Black dots are locations used for geoGAM calibration, locations with red triangles were used for model validation, ECEC legend classes according to Walthert et al., 2004).

BSS (30 cm: 0.332, 100 cm: 0.688), meaning that a penalized tensor product of spatial coordinates was too smooth to capture short range autocorrelation.

### 4.2.2  Validation of predicted presence of waterlogged horizons with independent data

Table 4 reports contingency tables for predicted outcomes for *presence of waterlogged horizons* at 198 sites of the validation set.

5  BSS and bias ratio improved again from the 0–30 cm to the 0-100 cm layer. In 0–30 cm depth *presence of waterlogged horizons* were clearly over-predicted and down to 50 cm slightly over-predicted while down to 100 cm there was no bias. Performance evaluated by percentage correct with the Peirce skill score (PSS) was similar for all three depths (correct predictions being 44 to 50 % more frequent compared to random predictions). Ignoring correct *absence* predictions in Gilbert skill score (GSS), the





**Table 4.** Observed occurrence of waterlogged horizons at three soil depths against predictions by geoGAM for the 198 sites of the validation set. Waterlogged soil horizons were predicted to be present if prediction probabilities were larger than an optimal threshold (30 cm: 0.22, 50 cm: 0.35, 100 cm: 0.51) found by cross-validation with GSS as criteria (#: number of sites per response level, BSS: Brier skill score, bias: bias ratio, PSS: Peirce skill score, GSS: Gilbert Skill score).

| waterlogged down to | # predicted | # observed present | absent | BSS | bias | PSS | GSS |
|---|---|---|---|---|---|---|---|
| 30 cm | present | 16 | 27 | 0.312 | 1.720 | 0.484 | 0.227 |
|  | absent | 9 | 146 | | | | |
| 50 cm | present | 28 | 25 | 0.448 | 1.152 | 0.444 | 0.267 |
|  | absent | 18 | 127 | | | | |
| 100 cm | present | 43 | 22 | 0.526 | 1.000 | 0.496 | 0.330 |
|  | absent | 22 | 111 | | | | |

model predicted the correct level 20–30 % more often than a random prediction scheme. Again, GSS increased with depth and larger chance of waterlogging occurring.

### 4.2.3 Mapping of presence of waterlogged horizons

Presence of waterlogged horizons in 0–30 cm was predicted for 13.8 % of the area of study region Greifensee (Fig. 6). For 0–50 cm this share increased to 27.3 % and in nearly 40 % of the soils waterlogged horizons were present in 0–100 cm. Waterlogged horizons were mapped in upper soil layers mainly on the larger plains to the north and south of Lake Greifensee. Deeper horizons had waterlogging present mostly in local depressions and comparably smaller valley bottoms in the hilly uplands to the south of the study region.

### 4.3 Drainge classes - case study 3

### 4.3.1 Model for drainage classes

The models for the ordinal drainage class data contained about the same covariates as the models for *presence of waterlogged horizons* (Table 2). Most covariates had only very weak nonlinear effects (Fig. S7 in Supplement). Residual spatial autocorrelation was very weak with a short range (Fig. S2 in Supplement) suggesting that the variation was well captured by the geoGAM. 10-fold cross-validation resulted in a ranked probability skill score (RPSS) of 0.588.

### 4.3.2 Validation of predicted drainage classes with independent data

Table 5 reports the number of correctly classified and misclassified drainage class predictions for the validation set. False predictions were equally distributed above and below the diagonal, hence predictions were unbiased with a mean bias ratio close



**Table 5.** Frequency of drainage class levels and predictions of respective outcomes by geoGAM for the 198 sites of the validation set (#: number of sites per response level, RPSS: ranked probability skill score, bias: mean bias ratio, PSS: Peirce skill score, GS: Gerrity score for ordered responses).

| # predicted | # observed | | | RPSS | bias | PSS | GS |
|---|---|---|---|---|---|---|---|
| | well drained | moderately well drained | poorly drained | | | | |
| well drained | 129 | 9 | 9 | 0.458 | 0.985 | 0.477 | 0.523 |
| moderately well drained | 9 | 9 | 3 | | | | |
| poorly drained | 8 | 5 | 17 | | | | |

to 1. Distinguishing *moderately well drained* soils from the other two classes remained difficult as this class had been seldom observed. Overall, the model accuracy was satisfactory, with RPSS of 0.458 being only slightly smaller than cross-validation RPSS. Hence, the geoGAM was clearly better than predicting always the most abundant class *well drained*. Measured by PSS and Gerrity score (GS), the geoGAM was better than random predictions at every second site, for which predictions were
computed.

### 4.3.3    Mapping of drainage classes

Drainage classes were again predicted using a 20 m-grid (Fig. 7). 73.2 % of the area of the Greifensee region had *well drained* soils. *Poorly drained* soils were predicted for only 15.6 % of the area. The location of *poorly drained* soils coincides with *presence of waterlogged horizons* in the topsoil (0–30 cm, panel [a] in Fig. 6). The largest contiguous area of *poorly drained*
soils was predicted on accumulation plains at the lake inflow to the south of Lake Greifensee. The sites misclassified had TPI values indicating local depressions and had larger erosion accumulation potential (MRVBF) compared to correctly classified sites, thus predicting correct drainage classes in valley bottoms seems more difficult. Misclassified sites of the validation set had on average slightly larger clay and soil organic carbon contents in topsoil.

## 5    Discussion

### 5.1    Model building and covariate selection

The model building procedure efficiently selected for all responses parsimonious models with $p \leq 17$ covariates for all responses. This corresponds to only 5.8 % of the covariates considered for the effective cation exchange capacity (ECEC) modelling and to 1.4–2.8 % for modelling the binary and ordinal responses describing waterlogging.

The procedure was able to select meaningful covariates, which reveal the influence of soil forming factors for the response
variable, without any prior knowledge about the importance of a particular covariate. No pre-processing of covariates, such as dealing with with multi-collinearity by reducing the dimensionality of the covariate set, was necessary. Especially for terrain





covariates this is important. A variety of algorithms are available to calculate e.g. curvature or topographic wetness indices (TWI) which each likely produce slightly different results. In addition, radii for computing e.g. topographic position indices (TPI) have to be specified and it is often not a priori clear how these should be chosen (Behrens et al., 2010; Miller et al., 2015). Therefore, different algorithm and a range of parameter values are used to create terrain covariates and the model

building process selects the most suitable input to model a particular soil property. Meanwhile, none of the 180 APEX bands available for the Greifensee region was chosen for the final models. Most likely, meaningful preprocessing – e.g. based on bare soil areas – could improve the usefulness of such covariates (Diek et al., 2016). Since we used continuous reflectance signals, including vegetated and sparsely vegetated areas, the remotely sensed signal might not have expressed too well direct relationships to actual soil properties.

## 5.2    Model structure

Parsimonious models lend themselves to a verification of fitted effects from a pedological perspective. Yet, due to multi-collinearities in the covariate set, effects of selected covariates could be substituted by effects of other covariates (Behrens et al., 2014).

Although Johnson et al. (2000) did not find strong relationships between terrain and ECEC, six terrain attributes were
selected. Covariates representing geology were important, too, with e.g. changing ECEC as a function of the distance to two types of moraines. Also, vegetation provided information on ECEC in the topsoil as a vegetation index (difference of near infrared to red reflectance) and a vegetation map were included. Larger values of ECEC were modelled for plant communities that are characteristic for nutrient-rich soils. The factor distinguishing the origin of soil data either from direct measurement or pedotransfer function (PTF, legacy data correction, Sect. 3.2.2, Fig. S1 in Supplement) was further relevant in the ECEC
model.

For modelling drainage classes and *presence of waterlogged horizons* plausible covariates were selected (Figs. S4 to S7 in Supplement). Most covariates were terrain attributes derived from the digital elevation model (DEM). This is in accordance with Campling et al. (2002) who found topography important in general and Lemercier et al. (2012) who showed that a topographic wetness index was among the most important covariates. Local depression at various scales (concave curvature, basins in TPI,
sites with accumulation by erosion, increased terrain wetness) increased the probability for *poorly drained* soils and *presence of waterlogged horizons*. More variable terrain (standard deviation of elevation) also increased waterlogging probability. Climate covariates also seemed to be important. Rainfall pattern in summer (June, July), spring dew point temperature and global radiation (March, April) correlated most strongly with *presence of waterlogged horizons*. Information on human activities related to waterlogged soil amelioration were included in all four models. Maps of historic wetlands and areas with drainage
systems were most often chosen in combination. Geology was also partly relevant (*presence of waterlogged horizons* in 0–100 cm soil depth and drainage classes).

Overall, nonlinearities in effects were small for drainage classes and *presence of waterlogged horizons*. Estimated degrees of freedom (EDF, Wood, 2006, pp. 170) were generally smaller than 1.5, with some continuous effects even being close to 1 EDF. In contrast, most nonlinear effects of the model for ECEC had EDF around 1.7–1.8 with northness consuming even 2.0




EDF. The large area of the study region and the response being a chemical property that depends on various combinations of soil forming factors evidently required the use of a more complex model.

### 5.3 Predictive performance of fitted models

In general, no over-fitting of the calibration data was observed. Cross-validation statistics of the final models were similar to

results obtained by independent validation. Independently validated model accuracy was satisfactory for ECEC in the present study with ($SS_{mse}$ 0.37). Building a separate model for forest soil ECEC for a dataset with about 2.1 sites per $km^2$ seem to produce much better results than the study reported by Vaysse and Lagacherie (2015) who found very poor model performance for ECEC ($R^2 = 0$, equivalently computed as $SS_{mse}$) for a dataset with 0.04 sites per $km^2$ and a study region with multiple land uses.

Presented models reached similar accuracy as reported in other studies. Zhao et al. (2013, Table 1) reported that 64 to 87 % of the sites were correctly classified (percentage correct, PC) in four studies that modelled three drainage class levels. Three studies with up to seven drainage levels achieved PC of 52 to 78% and Zhao et al. (2013) themselves had 36 % of correctly classified sites. Kidd et al. (2014) found PC of 53 % and 55 % for two study regions, and Lemercier et al. (2012) reported PC of 52 % for a four-level drainage response. The presented models (Table 4 and 5) are about as good with PC of 78 % to 82 %

for predicting *presence of waterlogged horizons* and PC of 78 % for predicting the three drainage class levels.

Nevertheless, PC is trivial to *hedge* (Jolliffe and Stephenson, 2012, pp. 46), and comparisons should be made only with care. Better performance measures are PSS and Cohen's kappa ($\kappa$), also called Heidke skill score (Wilks, 2011, pp. 347). Campling et al. (2002) reported a $\kappa$ of 0.705, Kidd et al. (2014) $\kappa$'s of 0.27 and 0.31 for the two study regions, Lemercier et al. (2012) a $\kappa$ of 0.27 and Peng et al. (2003) found $\kappa$ of 0.59 for predictions of three drainage levels. $\kappa$'s computed for the models of

this study ranged between 0.37 and 0.5 for modelling the *presence of waterlogged horizons* and was 0.48 for predicting the three levels of drainage class. Unequal distribution of the three drainage classes in the study region (majority of soils were *well drained*) were reflected in the smaller value of $\kappa$ compared to PC.





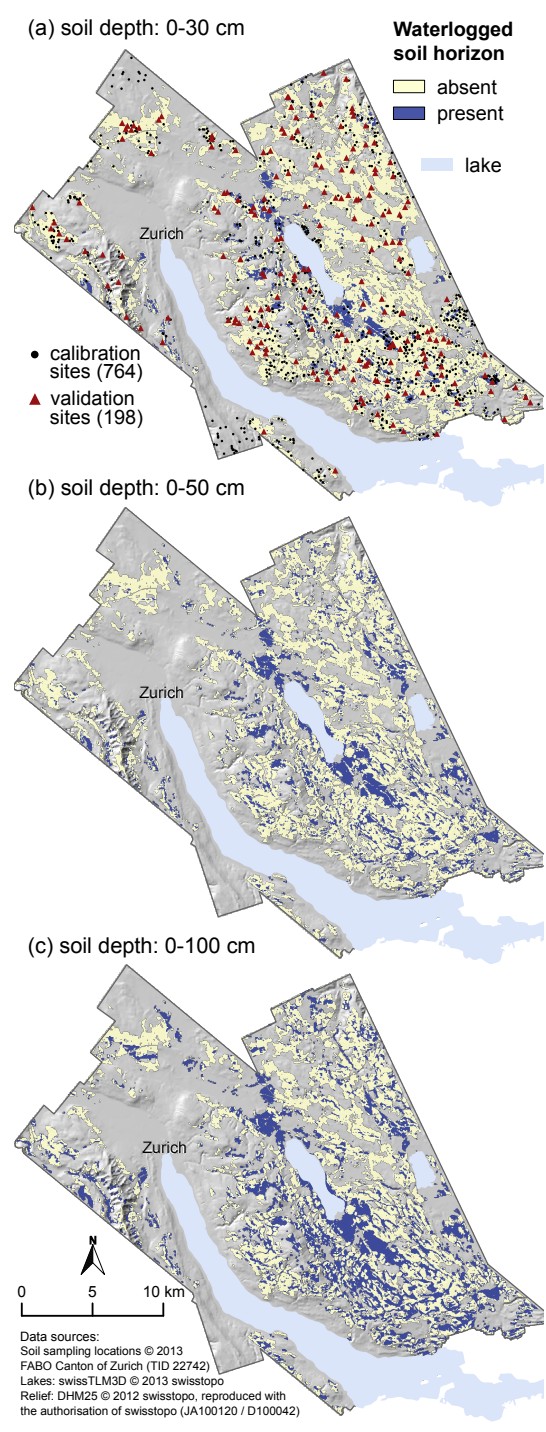

**Figure 6.** geoGAM predictions of presence of waterlogged horizons between surface and 3 soil depths (a: 0–30, b: 0–50, c: 0–100 cm) for the agricultural land in the Greifensee study region (computed on a 20 m-grid with final geoGAM with covariates according to Table 2, smoothed for better display with focal mean with radius of 3 pixels = 60 m). Black dots in panel (a) are locations used for geoGAM calibration, locations with red triangles were used for model validation.





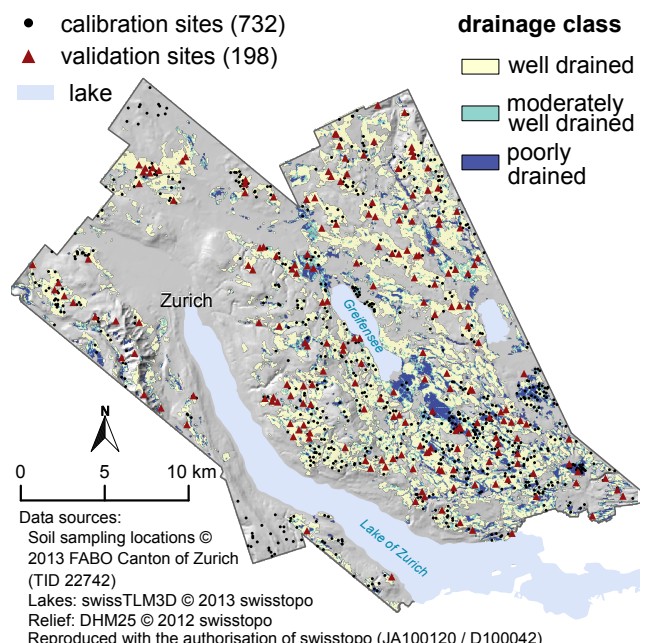

**Figure 7.** geoGAM predictions of drainage classes for the agricultural land in the Greifensee study region (computed on a 20 m-grid with final geoGAM with covariates according to Table 2, smoothed for better display with focal mean with radius of 3 pixels = 60 m). Black dots are locations used for geoGAM calibration, locations with red triangles were used for model validation.





## 5.4 Spatial structure of predicted maps

The spatial distribution of ECEC as shown by Fig. 5 aligns well with pedological knowledge about soils in the Canton of Zurich. The smallest ECEC ($< 50 \, \mathrm{mmol_c \, kg^{-1}}$) was mapped in the northeast of the study region. The last glaciation (Swisstopo, 2009) did not reach as far north and, as a consequence, strongly weathered soils on old fluvioglacial gravel-rich sediments developed in this part of the study region. Soils not covered by ice during the last glaciation have comparably larger ECEC if they formed on Molasse.

As expected the spatial patterns for the *presence of waterlogged soil horizons* and the drainage classes were very similar (Fig. 6 and 7). Especially soils on plains to the north and south of Lake Greifensee are often *poorly drained*, although at many locations agricultural drainage networks were installed in the past.

## 6 Summary and conclusion

Effectively building predictive models for digital soil mapping (DSM) becomes crucial if many soil properties are to be mapped. Selecting only a small set of relevant covariates renders interpretation of the fitted models easier and allows to check whether modelled relations accord with pedological understanding. Parsimonious, interpretable DSM models are likely more readily accepted by end-users than complex black-box models. Moreover, model selection out of a large number of covariates describing soil forming factors helps to improve knowledge about relationships at larger scales. In this sense, it is also important, that the model approach provides information about covariates which are not relevant for a certain response, e.g. the large number of APEX bands for *persence of waterlogged horizons* and drainage classes.

We developed a model building framework for generalized additive models for spatial data (geoGAM) and applied the framework to legacy soil data from the Canton of Zurich (Switzerland). We found that geoGAM

- consistently modelled continuous, binary and ordinal responses, hence, allow DSM of measured soil properties and soil classification data using one common approach,

- selected, given the large numbers of covariates, adequately small sets of pedogenetically meaningful covariates without any prior knowledge about their importance and without prior reduction of the covariate sets,

- required minimal user interaction for model building, which facilitates future map updates as new soil data or new covariates become available,

- allowed easy interpretation of effects of the included covariates by partial residual plots,

- modelled predictive distributions for continuous responses by a bootstrapping approach, thereby taking uncertainty of model building into account,

- did not over-fit the calibration data in our applications, and

 

– predicted soil properties with similar precision than other approaches did in other digital soil mapping studies, when tested with an independent validation set.

To further assess usefulness of geoGAM for DSM future work should focus on comparisons of predictive precision with commonly used statistical methods (e.g. geostatistics or tree-based machine learning techniques) on the same soil datasets.

5 *Code availability.* The geoGAM model building procedure was published as R package `geoGAM` (Nussbaum, 2017).

*Data availability.* The soil data were used under a non-public data licence (Canton of Zurich, contract number TID 22742; WSL) and could not be published.

*Author contributions.* A. Papritz proposed the application of componentwise gradient boosting with smooth baselearners for DSM. M. Nussbaum implemented the framework and adapted it to the needs of DSM. L. Walthert harmonized the soil data with collaborators, and M. 10 Fraefel computed multi-scale terrain attributes. L. Greiner defined the responses *presence of waterlogged soil horizons* and *drainage classes* from Swiss soil classification data. M. Nussbaum prepared the manuscript with major input from A. Papritz and further contributions from all co-authors.

*Competing interests.* The authors declare that they have no conflict of interest.

*Acknowledgements.* We thank the Swiss National Science Foundation SNSF for funding this work in the frame of the National Research 15 Program "Sustainable use of Soil as a Resource" (NRP 68)" and "Swiss Earth Observatory Network" (SEON) for funding aerial surveys with APEX. Special thanks go to WSL and the soil protection agency of the Canton of Zurich for sharing their soil data with us. Furthermore, we would like to thank Thorsten Hothorn for advice on model selection and boosting.



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
