# Peer review of "Mapping of soil properties at high resolution in Switzerland using boosted geoadditive models"

_SOIL, 2017_

## Referee Comment (RC1) · Anonymous Referee #1 · 3 Jul 2017

I. General comments This study makes an important methodological contribution to the Digital Soil Mapping research community by evaluating in depth a novel modelling technique (i.e. boosted geoadditive model), which clearly aims to find a good balance between the model predictive performance and its interpretability (as driven by the level of complexity). This is in particular useful, when a (very) large set of co-variates are / can be considered, and given the recent remarkable increase in data-resources availability, due to for example improved remote sensing techniques, the present model, will most probably have a large potential to be used in future research in soil science. Hence, I'm in favour of accepting this paper for publication in 'SOIL' journal (after minor revisions). Nevertheless, I think that some further clarifications are needed in terms of

the methodological approach. More precisely, it would be good to explain why both a K-fold cross validation as well as a validation based on an independent set of data were needed. Furthermore, I believe that the results needs to be discussed much more in detail, by comparing this study's output with other models' performances when predicting & mapping the considered variables in the literature. Please, consider as well my specific comments below (which will be helpful to tackle these general comments).

II. Specific comments P.2 L 20-30: You refer quite a lot to McBratney et al. 2003's overview paper on DSM. Although, I'm convinced that this is a very important paper and you certainly need to mention it, I believe that it would be much better to integrate as well much more specific examples when refereeing to specific modelling techniques as well as more recent publications. P. 2 L 45-46: Add in specific references? P. 2 L 50-52: You say "Lately" but subsequently refers to McBratney et al. 2003, which is a fairly old reference by now, so please, add more recent and specific references. P.4 L 80 – 90: It's unclear to me when K-fold cross validation will be used and when an independent set of data will be considered for validation. Moreover, why a "10-fold" has been considered (and not something different than 10?) P. 6 L 12-13 You state that the accuracy of the coordinates in about 25m. Is this something you interpret that way (because records have been made in the field on topographical maps) or has this info been documented somewhere? P. 6 L 35-39: Why did you consider this additional 5% of data for which you needed a PTF to estimate ECEC? Furthermore, did you test the effect of having included this data on the overall outcome / model performance / uncertainty ect. . .? P. 6 L 40 So that's 21.7% of the data used for Validation. Why 21.7%? P. 6 L 67 So that's 20.6% of the data used for Validation. Why 20.6%? P. 7-8 Section 3.3.3 It's fairly hard for me to understand when a certain statistical measure (to test the model predictive performance ect. . .) have been used on (i) the calibration data set or (ii) in the context of the K-fold cross-validation or (iii) considering the independent set of validation data-points. (See related general comment). Anyway, I guess it would be good to clarify this further in the MS and probably improve as well the structure of this section of the text in the light of this comment. P. 9 L 31-37: It
would be good to compare these results with results from other models as presented in the literature (See related general comment). I'm aware this need to be done in "Section 5.3" (see below / ultimate comment) P. 9 L 50-55: By saying that "the model explained about 40% of the variance of the log-transformed and 37% of the variance of the original data", I'm wondering if you did make this statement by comparing your RMSE-value with the STDEV value in the validation dataset? If yes, was it done in a (K-fold) cross validation context and/or based on the independent validation dataset? Can you please clarify what you did to come up with this conclusion? P. 10 Table 2: you refer to "SD slope" and "SD elevation" ect... as being the standard deviation in local neighbourhood. Can you please clarify this a little bit more? Does I got it right that you did calculate the standard deviation based on an X.X raster window in order to obtain alternative measures for terrain/topographical complexity?? P. 10 Table 2: Can you specify the 'r' (raster-resolution, i.e. 2m or 25 m) value of the maps of the considered topographical variables? P.13 L 10 – 25 & L 48 – 63. In the interpretation of the results as regards the influence of topography on the modelled soil characteristics it will be crucial to mention the raster-resolution, i.e. 2m or 25 m, because different levels of topographical detail will represents different process, i.e. small scales irregularity within a field (e.g. reflecting local surface irregularities related to agricultural practices) visible at 2 m resolution versus larger scale topographical general slope signature (e.g. reflecting variability induced by soil erosion processes) visible at 25m resolution. p. 15 Section 5.3. I believe that the discussion of the predictive performance of the fitted models can be worked out much more in detail by comparing this study's output with other models' performances when predicting & mapping the considered variables in the literature. (See related general comment)

---

## Referee Comment (RC2) · Anonymous Referee #2 · 8 Aug 2017

The manuscript is well written and deals with an important issue in digital soil mapping : How to take advantage of the multitude of co-variates and avoid problems of auto-correlation and model over fitting. The authors developed an automated routine using GAM models. They illustrate this routine with examples from different soil properties on a continuous scale (ECEC), binary scale (water logging) and ordinal scale (drainage class). I appreciate the complexity of the data and the statistics, but I would still insist on some quantitative measures to demonstrate the advantage of their approach: i) the selected variables in Table 2 should be more precise and the reader should be able to judge the relative importance and eventual interaction terms, ii) ii) Page 222 lines 4-5 How can the reader evaluate the benefits of the current approach in terms of reducing

the risk of overfitting? iii) The case of the APEX data illustrates that some co-variates are not clearly described and one could even discuss their use in the model. Page 10 line 16: It is well-known that spectral information depends on the development stage of the vegetation for crops and that grasslands are less sensitive to development stage. When were the data acquired and what was the hypothesis on the inference on water logging/drainage class from these spectra? Minor remarks Page 2 line 28 . . . .boosted regression trees. . . Page 3 line 14 You refer to review paper of 2003 to discuss the recent trends in the application of GAM's. Please check for some more recent papers. I am sure that GAM's have been applied to predict soil properties. Table 2 What does 'UK-DMC' mean?

---

## Author Comment (AC1) · 3 Sep 2017

Many thanks you for your helpful feedback. We comment on your review in the subsequent text (P: page, L: line of the manuscript in two-column layout). Please further consider our suggestions for changes of the manuscript in the supplement to this document.

[Figure]

**Cross-validation vs. validation with independent data (comments on P4 L 80–90, P7–8, Section 3.3.3)**

*More precisely, it would be good to explain why both a K-fold cross-validation as well as a validation based on an independent set of data were needed.*

*P.4 L 80–90: It's unclear to me when K-fold cross-validation will be used and when an independent set of data will be considered for validation. Moreover, why a "10-fold" has been considered (and not something different than 10)?*

The main goal for geoGAM modelling was computing predictions of soil properties. We therefore tuned the parameters of our model building procedure (e. g. number of boosting iterations, decisions on removal of covariates and merging of factor levels) based on cross-validation (cv) statistics evaluated with the calibration data set (and not based on "goodness-of-fit" measures computed with the same data). However, a model, built by repeatedly using cv, fits the calibration data "too closely", even if in a single application of cv, predictions and observations for a given subset are (seemingly) independent from each other. If we use cv repeatedly to tune parameters in a sequence of model building steps then predictions and observations of cv subset $k$ in step $i$ are no longer independent because the predictions depend on the observations of the $k$th subset through the parameters tuned before in steps 1 to $i-1$. cv prediction errors in steps $i > 2$ are likely too small, and the respective cv statistics are too optimistic compared to statistics evaluated for independent data. We can therefore consider cv statistics as conservative goodness-of-fit measures. Hence, to get an honest picture of the accuracy of predictions for new data, one should therefore use an independent validation data set.

For cross-validation Hastie et al. (2009, p. 254) recommend to use $k = 10$ subsets. Our data was spatially clustered. We also chose $k = 10$ to keep the $k$ calibration sets large enough to possibly still cover the full study area after the random splitting. With $k = 5$

some areas would most likely not any longer have been present in all $k$ calibration sets.

*P. 7–8 Section 3.3.3 It's fairly hard for me to understand when a certain statistical measure (to test the model predictive performance ect...) have been used on (i) the calibration data set or (ii) in the context of the K-fold cross-validation or (iii) considering the independent set of validation data-points. (See related general comment). Anyway, I guess it would be good to clarify this further in the MS and probably improve as well the structure of this section of the text in the light of this comment.*

We did not use the criteria introduced in section 3.3.3 as "goodness-of-fit" measures (i) for the calibration data set, but we used them as cross-validation statistics (ii) for model building, and as statistics to evaluate final model performance with independent validation data, see previous comment. We agree that the current text might be confusing and proposed clarifications (see supplement).

**Discussion of results (comments on P9 L31–37, P15, Section 5.3)**

*Furthermore, I believe that the results needs to be discussed much more in detail, by comparing this study's output with other models' performances when predicting + mapping the considered variables in the literature.*

*P. 9 L 31–37: It would be good to compare these results with results from other models as presented in the literature (See related general comment). I'm aware this need to be done in "Section 5.3" (see below / ultimate comment)*

*p. 15, Section 5.3. I believe that the discussion of the predictive performance of the fitted models can be worked out much more in detail by comparing this study's output with other models' performances when predicting + mapping the considered variables in the literature.*

As suggested we added more references on digital soil mapping of ECEC and discuss now the accuracy of the predictions achieved in the various studies (see section 5.3 in supplement). Moreover, we mentioned the results of Nussbaum et al. (2017) – a follow-up study focusing on a comparison of DSM methods for the same study regions and in the meantime published as a citable discussion paper.

We refrained from expanding the discussion on the accuracy of predictions for drainage class and presence of waterlogging. We cited a review that refers to 8 studies and 4 additional publications. Drainage classes were not often mapped, and their definition depends on local soil classification. Moreover, the validation statistics used in the various studies differ what makes comparison difficult. In the two last paragraphs of section 5.3 we nevertheless report for our study the statistics that were used by the other authors, although we think that these criteria are suboptimal.

**Answer to specific comments:**

References in introduction (P2 L20–30, L45–46, L50–52)

*You refer quite a lot to McBratney et al. 2003's overview paper on DSM. Although, I'm convinced that this is a very important paper and you certainly need to mention it, I believe that it would be much better to integrate as well much more specific examples when refereeing to specific modelling techniques as well as more recent publications. / P. 2 L 45-46: Add in specific references? / P. 2 L50-52: You say "Lately" but subsequently refers to McBratney et al. 2003, which is a fairly old reference by now, so please, add more recent and specific references.*

We agree on these comments and suggest to add more specific citations (see supplement).

Accuracy of coordinates (comment on P6 L12–13)

*You state that the accuracy of the coordinates in about 25 m. Is this something you in-
terpret that way (because records have been made in the field on topographical maps)
or has this info been documented somewhere?*

Unfortunately, we lack a thorough evaluation of the accuracy of geographical coordi-
nates. During the surveys, site coordinates were recorded on topographical maps at
a scale 1:25 000 without using positioning devices such as GPS or conventional land
survey instruments. From the topographical map scale (the thickness of a pencil line
(0.5–1 mm) corresponds to 12–25 m in the field) and by consulting soil surveyors we
estimated possible deviations to about 25 m. By reporting this figure, we wanted to
point out that the spatial resolution of some covariates was better than the accuracy of
the spatial coordinates of sampled locations. We propose to change the text slightly to
"we estimated accuracy of coordinates to about 25 m".

Pedotransferfunction for ECEC (comment on P6 L35–39)

*Why did you consider this additional 5 % of data for which you needed a PTF to esti-
mate ECEC? Furthermore, did you test the effect of having included this data on the
overall outcome / model performance / uncertainty ect...?*

We used these additional 73 sites (of which 13 were randomly assigned to the val-
idation set) because soil data was rather scarce. Furthermore, the bulk 95 % of the
data were not homogeneous either: Several institutions had gathered soil samples with
varying support (soil pits, fixed-depth aliquots collected on 20 x 20 $m^2$ plots) over about
30 years and had sent the samples for analysis to different labs. We could only partly
account for this variation (e. g. between-laboratory variation, Walthert et al. 2016) in

data harmonization. We did not evaluate how the use of data derived from PTF affected the uncertainty of the predictions. For such an analysis one would need to consider also other characteristics of legacy data (non-constant support, inter-laboratory variation, change of analytical methods, data coding schemes, . . . ) that all contribute to uncertainty.

Amount of data used for independent validation (comments on P6 L40, L67)

*P. 6 L 40 So that's 21.7 % of the data used for Validation. Why 21.7 %? P. 6 L 67 So that's 20.6 % of the data used for Validation. Why 20.6 %?*

We aimed to use about 20 % of the sites for model validation. After the splitting the data into calibration and validation sites we had to remove some sites because values were missing for some covariates (e. g. small clouds in SPOT5 satellite image). Missing values were more frequent in the calibration sets, hence the share of validation sites was eventually slightly larger than 20 %.

Explained variance (P9 L50–55)

*By saying that "the model explained about 40 % of the variance of the log-transformed and 37 % of the variance of the original data", I'm wondering if you did make this statement by comparing your RMSE-value with the STDEV value in the validation dataset? If yes, was it done in a (K-fold) cross-validation context and/or based on the independent validation dataset? Can you please clarify what you did to come up with this conclusion?*

With "explained variance" we referred to the mean squared error skill score (Eq. 14 in the manuscript) which is the ratio of the mean squared error and the variance of the

data. The reported "explained variances" all refer to the independent validation sets. We suggest to change the text to make this clear (see supplement).

Spatial standard deviation and raster resolution for topography (comments on P10, Table 2, P13 L 10–25, L 48–63)

*You refer to "SD slope" and "SD elevation" ect... as being the standard deviation in local neighbourhood. Can you please clarify this a little bit more? Does I got it right that you did calculate the standard deviation based on an X.X raster window in order to obtain alternative measures for terrain/topographical complexity?*

This is correct. We calculated standard deviations in a circular moving window to obtain terrain roughness measures. Different radii represent terrain changes at multiple scales. We expanded the explanation in the table caption (see supplement).

*P. 10 Table 2: Can you specify the 'r' (raster-resolution, i.e. 2m or 25 m) value of the maps of the considered topographical variables?*

We proposed to additionally add the radii of local neighbourhoods used to calculate the terrain attributes (see supplement).

*P.13 L 10–25 + L 48–63. In the interpretation of the results as regards the influence of topography on the modelled soil characteristics it will be crucial to mention the raster-resolution, i.e. 2m or 25 m, because different levels of topographical detail will represents different process, i.e. small scales irregularity within a field (e.g. reflecting local surface irregularities related to agricultural practices) visible at 2 m resolution versus larger scale topographical general slope signature (e.g. reflecting variability induced by soil erosion processes) visible at 25m resolution.*

[Figure]

We agree that not just algorithms and radii of moving neighbourhoods used to compute the various terrain attributes are relevant, but also the raster resolution of the original elevation data. We proposed to change the manuscript accordingly (see supplement).

**References**

Hastie, T., Tibshirani, R., and Friedman, J.: The Elements of Statistical Learning; Data Mining, Inference and Prediction, Springer, New York, 2 edn., 2009.

Nussbaum, M., Spiess, K., Baltensweiler, A., Grob, U., Keller, A., Greiner, L., Schaepman, M., and Papritz: Evaluation of digital soil mapping approaches with large sets of environmental covariates, SOIL Discussions, 2017, 1–32, 10.5194/soil-2017-14, http://www.soil-discuss.net/soil-2017-14/, 2017.

Walthert, L., Bridler, L., Keller, A., Lussi, M., and Grob, U.: Harmonisierung von Bodendaten im Projekt "Predictive mapping of soil properties for the evaluation of soil functions at regional scale (PMSoil)" des Nationalen Forschungsprogramms Boden (NFP 68), Bericht, Eidgenössische Forschungsanstalt WSL und Agroscope Reckenholz, Birmensdorf und Zürich, 10.3929/ethz-a-010801994, 2016.

Please also note the supplement to this comment:
https://www.soil-discuss.net/soil-2017-13/soil-2017-13-AC1-supplement.pdf

———————————————————

---

## Author Comment (AC2) · 3 Sep 2017

Many thanks for your helpful feedback. We comment on your review in the subsequent text (P: page, L: line of the manuscript in two-column layout). Please further consider our suggestions for changes of the manuscript in the supplement to this document.

[Figure]

**Details on covariate effects (Table 2)**

*The selected variables in Table 2 should be more precise and the reader should be able to judge the relative importance and eventual interaction terms.*

Complying with similar requests by referee 1, we added further details on covariates (radii of local windows used for computing terrain attributes, spatial resolution, etc.) to Table 2. Adding even more details — e. g. on non-linear effects of covariates — and keeping the table at the same time still well organized would be difficult. In the caption of Table 2 we refer to the partial residual plots in the Supplement of the manuscript where covariate effects are displayed. To keep the final models simple we did not include any interaction effects, which becomes apparent from the partial residual plots in Figures S1 and S4 to S7 in the Supplement.

Relative importance of covariates cannot be easily established for additive models (GAM) as opposed to tree-based methods. Estimated coefficients of one covariate depend on other covariates in the model. Moreover, because of collinearity, some covariates may be replaced by others without much loss of accuracy. To report covariate importance one could resort to a (disputed) partitioning of the coefficient of determination $R^2$ (Groemping 2006). We refrained from applying an ad-hoc method to evaluate covariate importance.

**Evaluation of overfitting**

*Page 22 lines 4-5 (P15 L2 in two-column manuscript): How can the reader evaluate the benefits of the current approach in terms of reducing the risk of overfitting?*

We used differences of statistics characterizing the accuracy of predictions for crossvalidation and independent validation as indicators for over-fitting. In view of a referee's comments on Nussbaum et al. (2017), we acknowledge that our use of the term "over-fitting" is not in accordance with its strict definition by Hastie et al. (2009). We therefore propose to clarify this at the begin of section 5.3 by replacing the first two sentences by the following text:

"For the final models, cross-validation statistics were similar to results obtained for the independent validation data. Through repeated cross-validation on the same subsets the cross-validation statistics are considered as conservative *goodness-of-fit* statistics (see our answer to a comment by referee 1). Hence, we conclude that the model did not over-fit the calibration data."

**APEX spectral data as covariates**

*The case of the APEX data illustrates that some co-variates are not clearly described and one could even discuss their use in the model. Page 10 line 16 (P6 L96 in two-column manuscript): It is well-known that spectral information depends on the development stage of the vegetation for crops and that grasslands are less sensitive to development stage. When were the data acquired and what was the hypothesis on the inference on water logging/drainage class from these spectra?*

Firstly, we agree that the information on covariates is rather sparse in the manuscript. Nevertheless, we suggest not to expand the manuscript with details on APEX imagery. The manuscript focuses on the geoGAM framework, hence we tried to avoid adding too much information specific to the study regions. As you well noted APEX bands were not relevant covariates, therefore we only included citations, to the benefit of readers interested in details, but did not go into details. According to Schaepmann et al. (2015) the flight campaigns took place in autumn and spring (09/2013, 04/2014) and captured

the vegetation in different stages. Furthermore, there were bare soil areas and soil moisture likely differed on the two occasions. Table S2 in the Supplement of Nussbaum et al. (2017) — a follow-up study focusing on a comparison of DSM methods for the same study regions and in the meantime published as a citable discussion paper — explains that we used an indicator covariate to account for different sampling dates. However, we likely could not fully correct discrepancies between the two flight campaigns, and this may be one reason that the APEX covariates were not selected for the final models.

Secondly, we also agree that the usage of certain covariates can be questioned. We did not expect to find a relationship between APEX spectral bands and the drainage class or presence/absence of waterlogging. But we included the covariates for two reasons: (i) We presumed that a priori exclusion of covariates based on expert knowledge would in general hamper accuracy of predictions (see e.g. Brungard et al. 2015). (ii) An automatic model building procedure should be capable to exclude non-relevant covariates. Indeed, none of the APEX bands was included in the final geoGAM models for these responses.

**Minor remarks**

We added clarifications and references as proposed (see supplement).

**Supplement:**

[revised manuscript text omitted]

---

## Author Response (AR1)

Eidgenössische Technische Hochschule Zürich
Swiss Federal Institute of Technology Zurich

**Department of Environmental System D-USYS**
**Soil and Terrestrial Environmental Physics STEP**

ETH Zurich
Ms. Madlene Nussbaum
PhD student
Universitätsstrasse 16
8092 Zurich, Switzerland

madlene.nussbaum@env.ethz.ch
www.step.ethz.ch

Prof. Dr.
Bas van Wesemael
Handling Topical Editor
Journal SOIL

Zurich, 11 September 2017

**Submission after minor revisions of manuscript No soil-2017-13**

Dear Editor

Thank you for your positive answer to our manuscript named *"Mapping of soil properties at high resolution in Switzerland using boosted geoadditive models"* (soil-2017-13).

We are pleased to resubmit the revised article. Point by point answers to the comments of the referees can be found in the public discussion of the manuscript. We refrained from repeating these answers here. After careful re-reading of the manuscript we further added small changes to ensure better readability (see attached list). Enclosed you find the marked-up version of the manuscript detailing the changes.

Many thanks for all your efforts to handle our manuscript.

Yours sincerely,

Madlene Nussbaum

Document with changes (latexdiff)

**Submission after minor revisions of manuscript No soil-2017-13**

Besides the proposed changes (see answers to referee 1 and 2) we revised the manuscript as follows (P: page, L: line of two-column discussion manuscript, first submission):

- P1, Abstract L3: removed "and discontinuous"
- P1, Abstract L5: removed "data"
- P2 L7: replaced "defined" by "specific"
- P2 L12: removed "discontinuous and"
- P2 L41: defined abbreviation lasso here (before on P4 L15)
- P2 L69: removed "of a case study"
- P2 L98: removed "studies"
- P2 L104: added "EDK, "
- P3 L18: replaced "optimizes" by "meets"
- P3 L22: replaced "we apply the method to three DSM case studies from the Canton.." by "we use the method in three DSM case studies in the Canton .."
- P3 L41: added the transpose
- P4 L4: replaced "number" by "seqence"
- P4 L32: replaced "with" by "by"
- P4 L36-38: harmonised notation to f(..) instead of g(..)
- P4 L67-77: revised begin of third step for better readability
- P4 L81, L101: replaced "to" by "by"
- P4 L84-85: removed "main" and ", that was fitted without offset."
- P4 L89: replaced "factors" by "terms"
- P5 L5: removed "fit"
- P5 L27: replaced "a" by "the"
- P5 L38: replaced ".. region was chosen near the Lake Greifensee by .." by "region near the Lake Greifensee was defined by.."
- P5 L47: "and in the"
- P5 L61-62: replaced "Jura foothills with limestone rocks" by "limestone Jura hills" (following review of soil-2017-14)
- P5 L74-75: replaced "purposively" by "by purposive sampling (Webster and Lark, 2013, p. 86) (following review of soil-2017-14)
- P5 L77 – P6 L1-2: replaced "Collating these soil data from different sources implicated that soil data were not directly comparable" by "Soil data was therefore quite heterogeneous"
- P6 L4: removed "of soil legacy data" and "among others"
- P6 L9: replaced "performance" by "the main"
- P6 L20-21: replaced "At about 200 sites soil profiles were sampled at genetic horizons." by "For about 100 sites soil profiles genetic horizons were sampled."
- P6 L28: removed sentence "The w_i were normalized to sum to 1."
- P6 L88: corrected "data in the covariates" to "supplementary qualifiers"
- P6 L95: replace "describing" by "to describe"
- P7 L13: Used s+ in notation to indicate predictions at new locations (as in section 2.3)
- Table 1: replaced "presence of aquifer" by "information on aquifers"
- P7 68-69, P8 L14-15: harmonised notation
- P8 L19-20: removed "the mean bias ratio and two skill scores: We calculated the"
- P8 L25: added "In addition,"
- P10 L2: replaced "probabilities" by "probability"
- P11 L16: removed "over-predicted"
- P12 L6 – P13 L1: remove "for all responses"
- P13 L6: replaced "for" by "on"
- P13 L8-11: Restructure sentence: "No pre-processing of covariates was necessary, e.g. such as reducing the dimensionality of the covariate set to deal with multi-collinearity."

**Submission after minor revisions of manuscript No soil-2017-13**

- P13 L20: replaced "input" by "covariates"
- P13 L38: replaced "changing ECEC" by "ECEC changing"
- P13 L40: replaced "as" by "because"
- P13 L55-59: We refrained from adding the proposed changes because this would hamper readability and stress on details not relevant for this section. This information is included in Table 2 and the Supplement.
- P15 L66: replaced "model" by "modelling"
- P15 L92: replaced "than" by "as"
- P15 L99: Added "Nussbaum et al. (2017) published a first such study."
- Following the review of soil-2017-14: replaced "precision" by "accuracy" throughout the manuscript
- Following the review of soil-2017-14: replaced "layer" by (soil) "depth"

[revised manuscript text omitted]